# Neo-Adjuvant Chemotherapy in Gastric Adenocarcinoma: Impact on Surgical and Oncological Outcomes in a Western Referral Center

**DOI:** 10.3390/cancers17152465

**Published:** 2025-07-25

**Authors:** Claudio Fiorillo, Beatrice Biffoni, Ludovica Di Cesare, Fausto Rosa, Sergio Alfieri, Lodovica Langellotti, Roberta Menghi, Vincenzo Tondolo, Giuseppe Quero

**Affiliations:** 1Digestive Surgery Unit, Fondazione Policlinico Universitario “Agostino Gemelli” IRCCS, Largo Agostino Gemelli 8, 00168 Rome, Italy; claudio.fiorillo@hotmail.it (C.F.); ludovica.dicesare@icloud.com (L.D.C.); fausto.rosa@policlinicogemelli.it (F.R.); sergio.alfieri@policlinicogemelli.it (S.A.); lodovica.langellotti2@guest.policlinicogemelli.it (L.L.); roberta.menghi@policlinicogemelli.it (R.M.); giuseppe.quero@policlinicogemelli.it (G.Q.); 2Department of translational medicine and surgery, Università Cattolica del Sacro Cuore di Roma, Largo Francesco Vito 1, 00168 Rome, Italy; vincenzo.tondolo@policlinicogemelli.it; 3General Surgery Unit, Fatebenefratelli Isola Tiberina–Gemelli Isola, Via di Ponte Quattro Capi 39, 00186 Rome, Italy

**Keywords:** gastric cancer, neo-adjuvant chemotherapy, gastrectomy, postoperative outcomes

## Abstract

Given the increasing use of neoadjuvant chemotherapy in the modern treatment of gastric cancer, this study aims to evaluate its real-world impact on perioperative and oncologic outcomes in a high-volume Western center. Based on our experience, neoadjuvant chemotherapy prior to gastrectomy does not appear to increase postoperative morbidity or mortality. However, in elderly or comorbid patients, its use should be carefully considered to identify the safest and most effective therapeutic strategy, with the goal of minimizing the risk of postoperative complications and the need for surgical reintervention.

## 1. Introduction

Gastric cancer (GC) remains a major cause of cancer-related mortality, with 1.1 million new cases and an estimated 770,000 deaths reported in 2020 [1]. In recent years, significant advances in multimodal treatment strategies have substantially improved the prognosis of patients affected by this disease [2]. While surgery has historically represented the cornerstone of GC treatment [3], the therapeutic landscape has evolved with the introduction of novel oncological therapies, particularly in the neo-adjuvant setting. In Western countries, an increasing number of GC patients are now candidates for neo-adjuvant chemotherapy (NACT) protocols, as endorsed by several international guidelines. This is emphasized in the European Society for Medical Oncology (ESMO) Guidelines published in 2022, which highlight the importance of a multimodal approach for patients with stage Ib–III disease (T > 1, Nx) [4]. Similar recommendations are provided by the 2024 National Comprehensive Cancer Network (NCCN) Guidelines.

Ref. [5] suggests that medically fit patients with potentially resectable local disease (cT2, Nx) may receive perioperative chemotherapy as an alternative to upfront surgery. In Italy, these indications are further supported by the Italian national guidelines (Associazione Italiana Oncologia Medica-AIOM and Gruppo Italiano Ricerca Carcinoma Gastrico-GIRCG) which recommend NACT for all GC patients staged as ≥T3 and/or N+ [6,7].

Although the current literature largely supports the efficacy of NACT in downstaging GC, eradicating micrometastases, improving treatment tolerance, increasing the likelihood of radical resection, and enhancing the R0 resection rate, the magnitude of the survival benefit remains debated. Some studies have demonstrated significant improvements in overall survival (OS) [8,9] while others have reported inconsistent survival advantages and a potential increase in perioperative complications associated with neo-adjuvant treatment [10,11]. In this context, various chemotherapy regimens have been proposed for GC [12,13], with the FLOT regimen currently considered the standard of care in Western countries. Alternative protocols are typically reserved for patients with contraindications or specific clinical conditions, as determined by oncologic assessment [4].

This study aimed to evaluate the safety and effectiveness of neo-adjuvant therapy regimens in a tertiary referral center for GC treatment and to assess whether neoadjuvant treatment—and the specific regimen used—may influence the postoperative course and long-term outcomes of patients undergoing gastrectomy for GC.

## 2. Materials and Methods

All patients aged ≥18 years who underwent surgery for GC between March 2016 and January 2024 at the Digestive Surgery Unit of the Fondazione Policlinico Universitario “Agostino Gemelli” IRCCS in Rome were retrospectively included in the study from a prospectively maintained database.

Demographic characteristics, comorbidities (including diabetes, cardiovascular, renal, and respiratory diseases), surgical details (type of resection and reconstruction technique), tumor features (size, location, histological subtype according to the Lauren classification [14], and pathological stage), the administration of neo-adjuvant therapy and type of neoadjuvant regimen, as well as postoperative course and follow-up data were retrospectively collected.

Tumor staging was classified according to the 8th Edition of the American Joint Committee on Cancer (AJCC) guidelines [15].

Post-operative complications were defined as surgery-related adverse events occurring within 30 days after the procedure and were graded according to the Clavien–Dindo classification [16].

All gastric resections were performed by the same surgical team using a standardized technique throughout the study period, as previously described [17,18]. An open surgical approach was adopted in all cases. For total gastrectomy (TG), a Roux-en-Y esophagojejunostomy was performed using a circular stapler, while for subtotal gastrectomy (STG), either a Billroth II or Roux-en-Y reconstruction was performed according to the surgeon’s preference. The length of the biliary limb in the Roux-en-Y reconstructions following TG or STG ranged between 40 and 60 cm. A D2 lymphadenectomy was performed in all cases.

The decision to administer NACT was made according to current GC treatment guidelines [10], following case-by-case discussions at a weekly multidisciplinary tumor board. Specifically, patients with suspected lymph node metastases or tumors staged as T ≥ 2 were considered candidates for neo-adjuvant therapy [4,6]. Conversely, obstructive or hemorrhagic GCs were deemed absolute contraindications to NACT [19]. Major comorbidities, particularly cardiovascular diseases, were considered relative contraindications and indication for NACT was based on an evaluation of the performance status and compliance of patients.

Only patients who completed the planned NACT regimen were included in the NACT group for analysis.

For long-term outcomes analysis, only patients who underwent surgery between March 2016 and December 2022 were included, ensuring a minimum follow-up period of two years.

### 2.1. Study Outcomes

The primary outcome of the study was to compare NACT and upfront surgery in terms of perioperative and long-term outcomes. Specifically, OS and disease-fee survival (DFS) were analyzed and compared between the two study groups. In addition, further analyses were conducted to identify factors influencing OS and DFS.

The secondary outcome was a comparison of perioperative outcomes within the NACT cohort based on the type of neoadjuvant regimen administered. In this subgroup, a multivariate analysis was performed to identify factors associated with the composite outcome of mortality and Clavien–Dindo ≥ 3 complications. A separate multivariate analysis was also conducted to determine independent risk factors for surgical reintervention.

### 2.2. Statistical Analysis

All continuous data were reported as median and quartile rank (QR) while numbers and percentages were used for all categorical data. Univariate analysis included Mann–Whitney U test, Student’s *t*-tests, χ^2^ test, and Fisher’s exact test. The survival analysis was conducted with the Kaplan–Meier method in order to estimate the long-term survival outcomes, and the log-rank test was used to analyze the statistical differences between the treatment groups. Multivariate analysis was performed with a multiple regression analysis, using the Cox proportional hazards model for survival analysis and using logistic regression for binary variables. The odds ratio (OR) for the long-term outcomes was determined with a Cox proportional hazards model. Statistical analysis was performed using commercially available software (SPSS^®^ for Windows version 25.0; Chicago, IL, USA). For all tests, a *p*-value ≤ 0.05 was considered statistically significant.

## 3. Results

### 3.1. Comparative Analysis of Clinico-Demographic and Perioperative Outcomes Between Upfront Surgery and NACT Patients

A total of 254 patients (162 males and 92 females) who underwent gastrectomy with D2 lymphadenectomy for GC between March 2016 and January 2024 at the Digestive Surgery Unit of the Fondazione Policlinico Universitario “Agostino Gemelli” IRCCS in Rome were included in the study. Of these, 144 patients (56.7%) did not receive neo-adjuvant therapy, and thus constituted the upfront surgery group, while 110 patients (43.3%) received neo-adjuvant therapy and represented the neoadjuvant chemotherapy (NACT) group.

Patients in the upfront surgery group were significantly older (72 (65–81) years vs. 65 (57–74) years; *p* ≥ 0.001) and had a higher prevalence of cardiovascular comorbidities (37–25.7% vs. 14–12.7%; *p* = 0.01) as compared to the NACT group.

Additionally, a higher ASA score and Charlson comorbidity index were evidenced in the upfront surgery group (56–38.9% vs. 15–13.6% *p* = 0.001; 125–86.8% vs. 72–65.5%; *p* = 0.001). Patients in the NACT group had a higher incidence of proximal tumors (*p* = 0.001) and more frequently underwent TG (54–49.1% vs. 44–30.6%; *p* = 0.003) and HIPEC (17–15.5% vs. 9–6.3%; *p* = 0.02) compared to those in the upfront surgery group. No statistically significant differences were found between the two groups in terms of length of hospital stay, postoperative mortality, or complication rates. Detailed clinico-demographic characteristics and perioperative outcomes are summarized in Table 1.

As shown in Table 2, patients in the NACT group more frequently presented lymph node metastases (82–74.5% vs. 84–58%; *p* = 0.007) and had a higher prevalence of diffuse histotype (27–45.8% vs. 39–31.7%; *p* = 0.03) compared to the upfront surgery group. Although the difference did not reach statistical significance, a trend toward a higher proportion of advanced T-stage tumors was observed in the NACT group (82–74.5% vs. 89–61.8% T3–4 stage; *p* = 0.07).

### 3.2. Long Terms Outcomes

The median follow-up for the whole cohort was 41 (1–75 months) months. At the last follow-up, 115 patients (45.3%) were alive, 56 (22%) were either lost to follow-up or underwent surgery after December 2022, and 83 (32.7%) had died of recurrence or other causes. No significant difference was observed in 5-year OS between the upfront surgery group (47.7%) and the NACT group (44.6%) (*p* = 0.96) (Figure 1).

An analysis of factors influencing 5-year DSF was performed. In the univariate analysis, patients younger than 65 years (*p* = 0.04), the presence of lymph node metastases (*p* = 0.001), advanced T stage (*p* = 0.001), positive surgical margins (*p* = 0.001), the presence of distant metastases (*p* = 0.001), and the administration of neo-adjuvant chemotherapy (*p* = 0.002) were identified as potential prognostic factors.

However, in the multivariate analysis using the Cox proportional hazard model, only the presence of lymph node metastases (OR: 2.5, C.I. 1.42–4.40; *p* = 0.001) and positive surgical margin status (OR: 1.89, C.I. 0.98–3.65; *p* = 0.006) were confirmed as independent negative prognostic factors for DFS.

### 3.3. Comparison Between FLOT Group and Other Regimes in Patients Who Underwent NACT

Overlooking the different types of chemotherapy regimens used (Appendix A), the majority of patients (74 out of 110–67.3%) received NACT according to the FLOT protocol (“FLOT group”), while the remaining 36 (32.7%) patients received a different neoadjuvant regimen (“non-FLOT group”).

A comparative analysis of the two subgroups revealed statistically significant differences. Patients in the FLOT group were generally younger (62 (60–67) years vs. 69 (65–73) years; *p* = 0.003) and had a lower ASA score ≥ 3 (6–7.9% vs. 9–26.5; *p* = 0.01) compared to those in the non-FLOT group.

With regard to intraoperative outcomes, patients in the FLOT group more frequently underwent TG (41–55.4% vs. 13–36.1%; *p* = 0.03), while the non-FLOT group showed a higher rate of associated organ resections (9–25% vs. 8–10.9% in the FLOT cohort; *p* = 0.05)

Postoperatively, neoadjuvant regimens other than FLOT were associated with a significantly higher rate of major complications (11–30.5% vs. 12–16.2%; *p* = 0.05), resulting in a higher need for surgical reintervention (8–22.2% vs. 4–5.4%, *p* = 0.008). No significant differences were observed between the two groups in terms of histopathological features. Clinico-demographic characteristics, perioperative outcomes and histopathological data are reported in Table 3 and Table 4.

### 3.4. Multivariate Analysis for Perioperative Mortality, Severe Complications, and Re-Intervention

A multivariate analysis was performed to identify predictors of the composite endpoint of postoperative mortality and severe complications (Clavien–Dindo grade ≥ 3). Tumor location emerged as the only independent risk factor (OR 4.7, C.I.: 1.56–14.18; *p* = 0.006) (Table 5).

Concerning the multivariate analysis for re-intervention, the use of a non-FLOT regimen and tumor location were identified as independent factors associated with an increased risk of surgical re-intervention (OR 0.22, C.I. 0.06–0.86; *p* = 0.003; OR 5.59, C.I. 1.1–28.45; *p* = 0.04) (Table 6).

## 4. Discussion

Over the last decade, the treatment of GC has evolved significantly. Although surgical resection remains the cornerstone of treatment of GC [20], its management has become progressively multimodal, with chemotherapy regimens playing a central role in improving patient prognosis [1,2,21].

Given the increasing adoption of NACT in this new era of GC treatment, we aimed to evaluate its impact on surgical and long-term outcomes based on our institutional experience.

The role of neo-adjuvant therapy gained increasing relevance following the publication of the MAGIC trial, which demonstrated the superiority of NACT over surgery alone in terms of OS and progression-free survival in patients with GC [21]. The MAGIC regimen was an anthracycline-based protocol consisting of epirubicin, cisplatin, and 5-fluorouracil (5-FU). This regimen remained the preferred option for nearly a decade until the results of the FLOT4 trial were published. The FLOT4 trial showed a significant improvement in DFS and OS, with acceptable toxicity, using a new combination regimen consisting of docetaxel, oxaliplatin, and 5-FU with leucovorin [22,23].

At present, based on the current evidence, the FLOT regimen is considered the standard NACT protocol for patients with gastric and gastroesophageal junction cancers.

Despite this, in clinical practice, the choice of chemotherapeutic agents is influenced by patient-specific factors, particularly the presence of cardiovascular or other comorbidities, which are commonly associated with an increased risk of treatment-related toxicity [24]. As outlined in the European Society of Cardiology (ESC) Guidelines on Cardio-Oncology, endorsed also by the Italian Society of Cardiology (SIC), different chemotherapeutic agents are associated with varying types and degrees of toxicity. Therefore, the selection of the most appropriate regimen used must be carefully individualized according to each patient’s clinical profile [25]. In this regard, a study by Hurria et al. developed a predictive model to identify patients at higher risk of chemotherapy-related toxicity, analyzing factors associated with the potential development of severe adverse events during NACT [26].

In line with these premises, the results of our study highlight significant differences in patient characteristics that likely influenced the decision to pursue either upfront surgery or NACT. Specifically, patients in the upfront surgery group were generally older (72 (65–81) vs. 65 (57–74); *p* = 0.001), had a higher prevalence of cardiovascular comorbidities (25.7% vs. 12.7; *p* = 0.01), and more frequently presented with higher Charlson comorbidity index scores (86.8% vs. 65.5%, *p* < 0.001). These findings can be attributed to the fact that patients with multiple comorbidities are often considered less suitable for chemotherapy given the increased risk of treatment-related toxicity [22].

Regarding tumor location and histological subtype, a higher incidence of proximal GCs and diffuse histotypes (predominantly located in the upper third of the stomach) was observed in the NACT group compared to the upfront surgery group (54–49.1% vs. 37–25.7%; *p* = 0.001 for proximal tumors; 27–45.8% vs. 39–31.7%; *p* = 0.03 for the diffuse histotype). Moreover, the NACT group presented with a more advanced stage, with lymph node metastases identified in 74.1% of cases, compared to 58% in the upfront surgery group (*p* = 0.007). As a result, a higher proportion of patients in the NACT group underwent total gastrectomy (49.1% vs. 30.6%, *p* = 0.003) and HIPEC (15.5% vs. 6.3%, *p* = 0.02). Notably, the combination of NACT and more extensive surgery in the subset of patients with a more advanced disease stage could achieve the same rate of R0 resection in the upfront surgery group (90–81.8% vs. 123–85.4%, *p* = 0.63). Regarding the extent of lymphadenectomy, a D2 dissection was performed in all cases, as recommended by current guidelines [27], with no significant difference in the number of lymph nodes harvested between the two groups (24 (14–45) vs. 26 (10–43); *p* = 0.91).

Looking at the potential impact of NACT on perioperative outcomes, comparable rates of severe complications and surgical reinterventions were observed in the two groups (18–17.15% vs. 17–12.8%; *p* = 0.54 and 12–10.9% vs. 24–16.7%; *p* = 0.19 in the NACT and upfront surgery groups, respectively). Similarly, no significant differences were found in the rates of anastomotic leakage (14–12.7% vs. 13–9.0%; *p* = 0.34) or in median postoperative hospital stay (10 (8–22) days vs. 9 (7–16) days; *p* = 0.26). These findings further support the perioperative safety of NACT and are consistent with the previously published literature [28,29,30,31].

In the subset of NACT patients, we further analyzed the potential impact of the different chemotherapy regimens on postoperative outcomes. Differing from the literature [30,31], higher rates of Clavien–Dindo with over three complications (11–30.5% vs. 12–16.2%; *p* = 0.05) and reintervention (8–22.2% vs. 4–5.4%; *p* = 0.008) were evidenced in the non-FLOT group. Moreover, non-FLOT regimens were independently associated with a higher risk of reintervention (OR: 0.22, 95% C.I.: 0.05–0.87; *p* = 0.003). Although the increased complication rate may be related to the differing biological effects of the chemotherapy regimens, as reported by Lorenzen et al. [31], we do believe these findings are more likely explained by patient selection. Indeed, older and more comorbid patients are less frequently eligible for FLOT due to its potential toxicity [32,33]. In our cohort, patients treated with regimens other than FLOT were significantly older (*p* = 0.003) and had a higher prevalence of ASA score ≥ 3 (*p* = 0.01), indicating a more fragile clinical profile. This interpretation is further supported by the propensity score-matched analysis conducted by Osama Moussa et al. [32], which showed no significant differences in postoperative complication rates between patients treated with FLOT and those treated with ECF/ECX when the groups were matched for age and comorbidities.

Although NACT is now widely used, its impact on long-term outcomes remains a matter of debate. Two meta-analyses have demonstrated that NACT in GC is associated with improved OS and DFS [34,35]. Conversely, Jian-Hong Yu et al. reported no clear prognostic benefit associated with NACT [11]. In our study, in contrast with the evidence reported by the MAGIC and FLOT4 trials [21,23], no significant difference was observed in 5-year OS between the NACT group (44.6%) and the upfront surgery group (47.7%) (*p* = 0.96). These findings may be partially explained by several limitations inherent to our study.

First, the relatively small sample size, the retrospective study design, and the presence of missing data (i.e., 72 cases with incomplete information on tumor histotype) significantly limit the generalizability of our results. Second, the lack of adequate matching between the two comparison groups in terms of clinico-demographic characteristics may have affected both perioperative and long-term outcome assessments; a particular potential limitation is that younger patients with advanced tumors may have received more aggressive treatments compared to older patients.

Differences in tumor location also represent an important limitation of this study. Notably, patients with gastroesophageal junction tumors often require more complex and higher-risk surgical procedures, which are inherently associated with an increased likelihood of postoperative complications. Third, the subset analysis within the cohort of NACT patients is limited by the few numbers of individuals for type of non-FLOT regimens, affecting the reliability of the outcomes obtained. Furthermore, since non-FLOT protocols involve a variety of chemotherapeutic agents with differing toxicity profiles and efficacy, grouping them into a single analytical category may have introduced significant bias into the analysis. On the other hand, our data reflect the real-world experience of a high-volume referral center for the surgical treatment of GC, offering a valuable contribution in supporting the safety and efficacy of NACT within multimodal treatment strategies for GC.

## 5. Conclusions

In conclusion, our study currently represents one of the few reports in the literature from a Western referral center for the treatment of GC, comparing perioperative and long-term outcomes between upfront surgery and NACT.

The findings of our study are in line with previously published data [28,30,31], likely supporting the safety of NACT in the treatment of GC, as it does not negatively impact hospital stay, complication rates, or perioperative mortality when compared to upfront surgery. Given the variability in patient frailty and individual capacity to tolerate NACT treatments, a comprehensive and personalized evaluation is essential to determine the safest and most appropriate therapeutic approach. In particular, the selection of the chemotherapy regimen must be tailored to each patient’s clinical profile. All cases should be discussed within a dedicated multidisciplinary tumor board to accurately assess eligibility and define the most suitable therapeutic approach. This collaborative decision-making model ensures optimal patient selection and promotes personalized treatment strategies, ultimately contributing to improved clinical outcomes.

Nevertheless, the considerable heterogeneity in chemotherapy regimens between the two cohorts, the relatively short follow-up period, and the substantial number of patients lost to follow-up prevent definitive conclusions from being drawn regarding the comparative long-term survival impact of NACT versus upfront surgery.

## Figures and Tables

**Figure 1 cancers-17-02465-f001:**
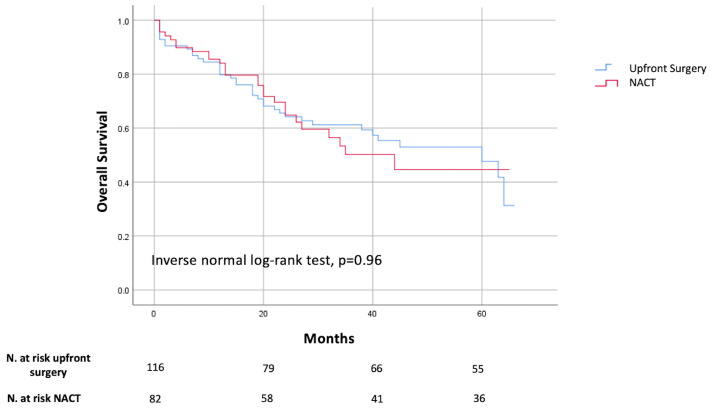
Kaplan–Meier curve comparing OS between upfront surgery and NACT.

**Table 1 cancers-17-02465-t001:** Clinico-demographic characteristics and perioperative outcomes of the study population.

	Study Population (n = 254)	Upfront Surgery (n = 144)	NACT (n = 110)	*p*
** *Clinico-demographic characteristics* **				
**Sex,** *n (%)*				
*Male*	162 (63.8)	84 (58.3)	78 (70.9)	0.04
*Female*	92 (36.2)	60 (41.7)	32 (29.1)	
**Age,** *years, median (QR)*	69 (61–78)	72 (65–81)	65 (57–74)	**0.001**
**Diabetes,** *n (%)*	38 (15)	23 (16)	15 (13.6)	0.60
**Cardiovascular diseases,** *n (%)*	51 (20.1)	37 (25.7)	14 (12.7)	**0.01**
**Respiratory diseases,** *n (%)*	21 (8.3)	11 (7.6)	10 (9.1)	0.67
**Nefropathy,** *n (%)*	3 (1.2)	3 (7.6)	0	0.13
**Smoker,** *n (%)*				
*Current*	27 (10.6)	13 (9)	14 (12.7)	0.05
*Ex*	67 (26.2)	31 (21.5)	36 (32.7)	
**ASA ≥ 3,** *n (%)*	71 (28)	56 (38.9)	15 (13.6)	**0.001**
**Charlson > 3,** *n (%)*	197 (77.6)	125 (86.8)	72 (65.5)	**0.001**
**Tumor location,** *n (%)*				
*Distal*	101 (39.8)	69 (47.9)	32 (29.1)	**0.001**
*Middle*	62 (24.4)	38 (26.4)	24 (21.8)	
*Proximal*	91 (35.8)	37 (25.7)	54 (49.1)	
** *Perioperative outcomes* **				
**Surgical procedure,** *n (%)*				
*Subtotal gastrectomy*	156 (61.4)	100 (69.4)	56 (50.9)	**0.003**
*Total gastrectomy*	98 (38.6)	44 (30.6)	54 (49.1)	
**HIPEC,** *n (%)*	26 (10.3)	9 (6.3)	17 (15.5)	**0.02**
**Associated organs resection,** *n (%)*	42 (16.5)	25 (17.4)	17 (15.5)	0.68
**Clavien–Dindo ≥ 3 complications,** *n (%)*	53 (20.9)	30 (20.8)	23 (20.9)	0.99
**Anastomotic leakage,** *n (%)*	27 (10.6)	13 (9)	14 (12.7)	0.34
**Reoperation,** *n (%)*	36 (14.2)	24 (16.7)	12 (10.9)	0.19
**Post-operative mortality,** *n (%)*	16 (6.3)	11 (7.6)	5 (4.5)	0.32
**LOS,** *days, median (QR)*	9 (7–15)	9 (7–16)	10 (8–22)	0.26

NACT: neoadjuvant chemotherapy; ASA: American Society of Anaesthesiologists; HIPEC: Hyperthermic Intraperitoneal Chemotherapy; LOS: length of stay.

**Table 2 cancers-17-02465-t002:** Histopathological features.

	Study Population (n = 254)	Upfront Surgery (n = 144)	NACT (n = 110)	*p*
**T stage,** *n (%)*				
*T0*	6 (2.4)	3 (2.1)	3 (2.8)	0.07
*T1–2*	77 (30.3)	52 (36.1)	25 (22.7)	
*T3–4*	171 (67.3)	89 (61.8)	82 (74.5)	
**N stage,** *n (%)*				
*N0*	88 (34.6)	60 (42)	28 (25.9)	**0.007**
*N+*	166 (65.4)	84 (58)	82 (74.1)	
**Harvested lymph nodes,** *median (QR)*	25 (14–40)	26 (10–43)	24 (14–45)	0.91
**Positive lymph nodes,** *median (QR)*	2 (0–10)	1 (0–10)	3 (0–12)	**0.01**
**Histotype *,** *n(%)*				
*Intestinal*	91 (35.9)	70 (56.9)	21 (35.6)	**0.03**
*Diffuse*	66 (25.9)	39 (31.7)	27 (45.8)	
*Mixed*	25 (9.9)	14 (11.4)	11 (18.6)	
**R status,** *n(%)*				
*0*	213 (83.9)	123 (85.4)	90 (81.8)	0.63
*1*	29 (11.4)	14 (9.7)	15 (13.7)	
*2*	12 (4.7)	7 (4.9)	5 (4.5)	

NACT: neoadjuvant chemotherapy; R: resection margin; * 72 missing data.

**Table 3 cancers-17-02465-t003:** Clinico-demographic characteristics and perioperative outcomes of the NACT group.

	Study Population (n = 110)	Non-FLOT (n = 36)	FLOT (n = 74)	*p*
** *Clinico-demographic characteristics* **				
**Sex,** *n (%)*				
*Male*	78 (70.9)	26 (78.8)	52 (68.4)	0.39
*Female*	32 (29.1)	8 (23.5)	24 (31.6)	
**Age,** *years, median (QR)*	65 (62–69)	69 (65–73)	62 (60–67)	**0.003**
**Diabetes,** *n (%)*	15 (13.6)	4 (11.8)	11 (14.5)	0.70
**Cardiovascular diseases,** *n (%)*	14 (12.7)	7 (20.6)	7 (9.2)	0.1
**Respiratory diseases,** *n (%)*	10 (9.1)	5 (14.7)	58 (6.6)	0.17
**Nefropathy,** *n (%)*	0			
**Smoker,** *n (%)*				
*Current*	14 (12.7)	3 (8.8)	11 (14.5)	0.22
*Ex*	36 (32.7)	15 (44.1)	21 (27.6)	
**ASA ≥ 3,** *n (%)*	15 (13.6)	9 (26.5)	6 (7.9)	**<0.01**
**Charlson > 3,** *n (%)*	72 (65.5)	26 (72.2)	46 (62.2)	0.29
**Tumor location,** *n (%)*				
*Distal*	32 (29.1)	11 (30.6)	21 (28.4)	0.66
*Middle*	24 (21.8)	6 (16.7)	18 (24.3)	
*Proximal*	54 (49.1)	19 (52.8)	35 (47.3)	
** *Perioperative outcomes* **				
**Surgical procedure,** *n (%)*				
*Subtotal gastrectomy*	56 (50.9)	23 (63.9)	33 (44.6)	**0.003**
*Total gastrectomy*	54 (49.1)	13 (36.1)	41 (55.4)	
**HIPEC,** *n (%)*	17 (15.5)	5 (11.8)	12 (17.1)	0.47
**Associated organs resection,** *n (%)*	17 (15.5)	9 (25)	8 (10.9)	**0.05**
**Clavien–Dindo ≥ 3 complications,** *n (%)*	23 (20.9)	11 (30.5)	12 (16.2)	**0.05**
**Anastomotic leakage,** *n (%)*	14 (12.7)	7 (19.4)	7 (9.6)	0.14
**Reoperation,** *n (%)*	12 (10.9)	8 (22.2)	4 (5.4)	0.008
**Post-operative mortality,** *n (%)*	5 (4.5)	3 (8.3)	2 (2.7)	0.12
**LOS,** *days, median (QR)*	10 (8–22)	12 (7–30)	9 (7–17)	0.18

FLOT: 5-fluorouracil, leucoverin, oxaliplatin, docetaxel; ASA: American Society of Anaesthesiologists; HIPEC: Hyperthermic Intraperitoneal Chemotherapy; LOS: length of stay.

**Table 4 cancers-17-02465-t004:** Histopathological features of the NACT group.

	Study Population (n = 110)	Non-FLOT (n = 36)	FLOT (n = 74)	*p*
**T stage,** *n (%)*				
*T0*	3 (2.7)	2 (5.6)	1 (1.4)	0.40
*T1–2*	25 (22.7)	7 (19.4)	18 (24.3)	
*T3–4*	82 (74.6)	27 (75)	55 (74.3)	
**N stage,** *n (%)*				
*N0*	28 (25.4)	11 (30.6)	17 (23)	0.39
*N+*	82 (74.6)	25 (69.4)	57 (77)	
**Harvested lymph nodes,** *median (QR)*	24 (14–45)	20 (9–53)	27 (15–44)	0.10
**Positive lymph nodes,** *median (QR)*	3 (0–12)	1 (0–12)	4 (0–13)	0.10
**Histotype *,** *n(%)*				
*Intestinal*	21 (19.1)	6 (40)	15 (34.1)	0.48
*Diffuse*	27 (24.5)	5 (33.3)	22 (50)	
*Mixed*	11 (10)	4 (26.7)	7 (15.9)	
**R status,** *n(%)*				
*0*	90 (81.8)	28 (77.8)	62 (83.8)	0.41
*1*	15 (13.6)	5 (13.9)	10 (13.5)	
*2*	5 (4.6)	3 (8.3)	2 (2.7)	

FLOT: 5-fluorouracil, leucoverin, oxaliplatin, docetaxel; R: resection margin. * 51 missing data

**Table 5 cancers-17-02465-t005:** Univariate and Multivariate analysis for Clavien–Dindo more than three complications in FLOT–no FLOT.

Univariate Analysis				
	Study Population	Clavien < 3	Clavien ≥ 3	*p*
**Age ≥ 65**	56	43 (76.8)	13 (23.2)	0.55
**Cardiovascular diseases,** *n (%)*	14	9 (64.3)	5 (35.7)	0.14
**ASA ≥ 3,** *n (%)*	15	9 (60)	6 (40)	**0.05**
**Charlson > 3,** *n (%)*	72	56 (77.8)	16 (22.2)	0.64
**Proximal tumor**	54	36 (66.7)	18 (33.3)	**0.002**
**Surgical procedure,** *n (%)*				
*Subtotal gastrectomy*	56	45 (80.4)	11 (19.6)	0.74
*Total gastrectomy*	54	42 (77.8)	12 (22.2)	
**HIPEC,** *n (%)*	17	16 (94.1)	1 (5.9)	0.09
**FLOT**	74	62 (83.8)	12 (16.2)	0.08
**Multivariate analysis**				
**Variables**	**OR**	**95% C.I.**	** *p* **	
**FLOT**	0.5	0.18–1.39	0.18	
**ASA ≥ 3**	1.80	0.50–6.44	0.36	
**Upper third location**	4.7	1.56–14.18	**0.006**	

See note 3 above.

**Table 6 cancers-17-02465-t006:** Univariate and multivariate analyses for reoperation in FLOT–non FLOT groups.

Univariate Analysis				
	Study Population	No-Reoperation	Reoperation	*p*
**Age** ≥ **65**	56	52 (92.8)	4 (7.4)	0.25
**Cardiovascular diseases,** *n (%)*	14	11 (78.6)	3 (21.4)	0.18
**ASA ≥ 3,** *n (%)*	15	11 (73.3)	4 (26.7)	**0.03**
**Charlson > 3,** *n (%)*	72	64 (88.9)	8 (11.1)	0.92
**Proximal tumor**	54	44 (81.5)	10 (18.5)	**0.01**
**Surgical procedure,** *n (%)*				
*Subtotal gastrectomy*	56	51 (91.1)	5 (8.9)	0.49
*Total gastrectomy*	54	47 (87)	7 (13)	
**HIPEC,** *n (%)*	17	16 (94.1)	1 (5.9)	0.47
**FLOT**	74	70 (94.6)	4 (5.4)	**0.008**
**Multivariate analysis**				
**Variables**	**O.R.**	**95% C.I.**	** *p* **	
**FLOT**	0.22	0.06–0.86	**0.003**	
**ASA ≥ 3**	1.85	0.41–8.35	0.43	
**Upper third location**	5.59	1.1–28.45	**0.04**	

See note 3 above.

## Data Availability

Data are available upon reasonable request at beatrice.biffoni@guest.policlinicogemelli.it.

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
