# Peer review of "Neo-Adjuvant Chemotherapy in Gastric Adenocarcinoma: Impact on Surgical and Oncological Outcomes in a Western Referral Center"

_cancers, 2025, doi:10.3390/cancers17152465_

Round 1
Reviewer 1 Report
Comments and Suggestions for Authors
- please clarify in the methods section in the abstract that the primary outcome was OS and DFS, please also do this in the methods section more clearly line 113-115 where the primary outcome is comparing between NACT and upfront surgery to assess OS/DFS, not in addition to
- I would also like the abstract to show this with the numbers, you mention that they were not different, without objective data here
- Your conclusion both in the abstract and at the end needs to be revised, Athough emphasizing individualized evaluation and multidisciplinary discussion, as you have done, is appropriate and consistent with current best practice, I would caution againt making a strong statement about the safety and generalizability of NACT for GC based on your study design. Especially since, these patients have higher rates of treatment discontinuation and are less likely to proceed to surgery after NACT. However, among those who complete therapy and undergo resection, perioperative complication rates and survival outcomes are generally comparable to younger or less comorbid patients (DOI: 10.1007/s10120-023-01404-2, DOI: 10.1245/s10434-023-14569-y)
- Please acknowledge this in the limitations. The NACT group was much younger and had fewer comorbidities with lower CCI between the two groups, which then introduces selection bias. Although multivariate analysis was performed, residual confounding is still likely. Moreover, this is also what likely resulted in the decision for partial vs total gastrectomy, the latter being more likely in the NACT group. As a result, the NACT group had higher rates of proximal tumors, subsequent total gastrectomies, and HIPEC use which is not a feature of the therapy itself, but rather reflects that surgeons are selecting younger, fitter patients with more advanced/complex disease for a more aggressive, multimodal treatment path. The conclusion that NACT is "safe" is confounded because it was given to a much safer, healthier cohort
- Please also include KPI or ECOG in this patient population, since in line 117, you mention how performance status was considered without objective data supporting this
- Please also acknowledge this in the introduction, tht while NACT improves R0 resection rates and may improve survival, the magnitude of survival benefit remains debated, with some studies showing inconsistent long-term survival advantages and a possible increase in perioperative complications (DOI: 10.1007/s11605-023-05641-9, doi: 10.1097/CM9.0000000000001603). I understand that you have discussed this in the discussion; however, you want to show why your study was important to shed light to this grey area
- Please list all the abbreviations used in the footnote of all the tables, like ASA, HIPEC etc.
- I would also acknowledge how no-FLOT group includes FOLFOX, ECF/ECX, cisplatin/5-FU, etc. all of which are different with vastly different toxicities and efficacies.
- Proximal tumors likely includes GEJ and true gastric cardia tumors which although treated similarly have different prognosis and either should be delinated or addressed as a limitation as potential heterogeneity
- lastly, The high percentage of missing data (72 missing data for tumor histo) must be explicitly addressed as a major limitation
Reviewer 2 Report
Comments and Suggestions for Authors
The present study explores the impact of neo-adjuvant chemotherapy on the outcomes after gastrectomies for cancer in a single-center experience from the Western world. The study is well-designed, the methods are correctly used, and the results support the conclusions. While the manuscript is well-written, the main concern of the paper is that it lacks novelty in the field. The safety and effectiveness of NAT in gastric cancer have been previously studied in other studies, including those involving a larger number of patients.
Author Response
Comments 1: The present study explores the impact of neo-adjuvant chemotherapy on the outcomes after gastrectomies for cancer in a single-center experience from the Western world. The study is well-designed, the methods are correctly used, and the results support the conclusions. While the manuscript is well-written, the main concern of the paper is that it lacks novelty in the field. The safety and effectiveness of NAT in gastric cancer have been previously studied in other studies, including those involving a larger number of patients.
Response 1: We thank the reviewer for the valuable comment, and we do appreciate the positive feedback. We agree that previous studies on this topic are already available in the literature. With our manuscript, we aimed to provide a further contribution to this topic by reporting a confirmatory experience from a high-volume referral center in the Western world.
Reviewer 3 Report
Comments and Suggestions for Authors
Thank you for this important study on neoadjuvant chemotherapy outcomes in gastric cancer. As a gastric cancer surgeon, I appreciate the institutional experience from Gemelli. However, there are significant concerns about the methodology that require major revision before publication.
This is the biggest issue with the study. The baseline characteristics show dramatic differences between groups that reflect clinical decision-making rather than random allocation. NACT patients were much younger (65 vs 72 years), healthier (fewer cardiovascular comorbidities, lower ASA scores), and had different tumor characteristics. This fundamental imbalance makes meaningful comparison impossible and undermines the conclusions.
The current presentation suggests these are comparable groups when they clearly aren't. From a surgical perspective, a 65-year-old with good performance status versus a 72-year-old with cardiovascular disease will have different baseline risks regardless of neoadjuvant therapy. The authors need to either use propensity score matching or acknowledge this as a major limitation that restricts conclusions to safety rather than effectiveness.
The lack of survival benefit (44.6% vs 47.7%, p=0.96) contradicts well-established evidence from MAGIC and FLOT4 trials. This needs substantial explanation. Were there issues with treatment completion? Patient selection? The discussion barely addresses this discrepancy, which is puzzling given the strong evidence for neoadjuvant benefits in gastric cancer.
From a surgical perspective, key information is missing that would help interpret the results. The authors should report treatment completion rates, reasons for discontinuation, nutritional status data (albumin, weight loss), performance status scores, and specific complications that led to reoperations. The finding that non-FLOT regimens led to more reoperations (22.2% vs 5.4%) is interesting but needs better explanation of what specific complications drove this difference.
The reassuring finding is the complication data showing similar rates of severe complications (20.9% vs 20.8%) and anastomotic leaks (12.7% vs 9.0%). This suggests that in appropriately selected patients, NACT doesn't significantly increase surgical risk, which matches clinical experience and is valuable information for practicing surgeons.
The authors need post-hoc power analysis for the negative survival findings to distinguish between "no effect" and "insufficient power." The 28% missing histotype data affects reliability of the Lauren classification analyses. The heterogeneous non-FLOT group (six different regimens plus "other") makes interpretation difficult. Short follow-up (41 months median) and 22% loss to follow-up introduce additional limitations.
The authors should reposition this as a surgical safety analysis demonstrating that NACT is safe in selected patients rather than a comparative effectiveness study. The focus should be on what the data actually supports - the safety message is valuable and believable, while the survival equivalence claims are problematic due to selection bias.
The authors should acknowledge the selection bias prominently in the abstract and discussion. They need to explain why the survival results differ from published trials. Treatment completion rates and nutritional data should be included. Kaplan-Meier survival curves should be added. The manuscript needs professional English editing throughout.
From a practical standpoint, what can clinicians tell patients based on this study? The safety data is reassuring for well-selected patients, but the survival data cannot be used to counsel patients about neoadjuvant benefits. The study is more useful as a safety analysis than an efficacy study.
This represents good institutional experience showing that NACT is surgically safe in appropriately selected patients. However, the survival conclusions are problematic due to selection bias. The paper would be stronger if repositioned as a surgical safety analysis rather than claiming treatment equivalence. With appropriate acknowledgment of limitations and focus on what the data actually demonstrates, this could make a valuable contribution to the literature. The manuscript needs major revision but has potential value for the surgical community.
Author Response
|
Comments 1: This is the biggest issue with the study. The baseline characteristics show dramatic differences between groups that reflect clinical decision-making rather than random allocation. NACT patients were much younger (65 vs 72 years), healthier (fewer cardiovascular comorbidities, lower ASA scores), and had different tumor characteristics. This fundamental imbalance makes meaningful comparison impossible and undermines the conclusions. The current presentation suggests these are comparable groups when they clearly aren't. From a surgical perspective, a 65-year-old with good performance status versus a 72-year-old with cardiovascular disease will have different baseline risks regardless of neoadjuvant therapy. The authors need to either use propensity score matching or acknowledge this as a major limitation that restricts conclusions to safety rather than effectiveness.
|
|
Response 1: We do appreciate this comment. We agree with the reviewer on the heterogeneity of the study populations and on the consequent lack of balance between the two comparative groups. Although this limitation has been already reported in the Discussion section, we further underlined this drawback adding the following paragraphs to the Discussion section (page 12, lines 334-342):
“…the lack of adequate matching between the two comparison groups in terms of clinico-demographic characteristics may have affected both perioperative and long-term outcome assessments; in particular, a potential limitation is that younger patients with advanced tumors may have received more aggressive treatments compared to older patients. Differences in tumor location also represent an important limitation of this study. In particular, patients with gastroesophageal junction tumors often require more complex and higher-risk surgical procedures, which are inherently associated with an increased likelihood of postoperative complications.”
Despite a propensity score matching could balance the difference between the two groups, the limited number of patients in our cohort did not allow for a statistically robust matching analysis. In order to partially overcome this limitation, we performed a multivariate analysis to adjust for these variables even if we acknowledge that residual confounding cannot be entirely excluded.
|
|
Comments 2: The lack of survival benefit (44.6% vs 47.7%, p=0.96) contradicts well-established evidence from MAGIC and FLOT4 trials. This needs substantial explanation. Were there issues with treatment completion? Patient selection? The discussion barely addresses this discrepancy, which is puzzling given the strong evidence for neoadjuvant benefits in gastric cancer.
Response 2: We do thank the reviewer for highlighting this point. We acknowledge that the absence of a survival benefit in our cohort appears to contrast with the well-established findings of the MAGIC and FLOT4 trials, which demonstrated improved survival with perioperative chemotherapy for GC. This discrepancy is likely multifactorial: the small simple size of the study population, the retrospective study design and the presence of missing data may have affected our results. In addition, the no-random selection of patients for NACT may have played a key role in obtaining these contrasting data. Indeed, there may have been a selection bias favoring younger and healthier patients for NACT. Similarly, the inclusion of more advanced GC and proximal and junctional tumors, that generally are indications to NACT and more invasive and high-risk surgical procedures such as total gastrectomy, may have significantly contributed to the contrasting findings regarding the long-term results. With regards to the completion rate of the NACT treatment, all patients completed the neoadjuvant treatment and the completion itself was an inclusion criterion. We do apologize not to have clearly reported this information in the Materials and Methods section. Please find the following statement added to the Materials and Methods section (page 3, lines 124-125): “Only patients who completed the planned NACT regimen were included in the NACT group for analysis.” In order to better underline the above-mentioned limitations, we modified the Discussion section as follows ( page 12, line 327-338):
“In our study, in contrast with the evidence reported by the MAGIC and FLOT4 trials[21,23], no significant difference was observed in 5-year OS between the NACT group (44.6%) and the upfront surgery group (47.7%) (p=0.96). These findings may be partially explained by several limitations inherent to our study. First, the relatively small sample size, and the retrospective study design and the presence of missing data (i.e. 72 cases with incomplete information on tumor histotype) significantly limit the generalizability of our results. Second, the lack of adequate matching between the two comparison groups in terms of clinico-demographic characteristics may have affected both perioperative and long-term outcome assessments; in particular, a potential limitation is that younger patients with advanced tumors may have received more aggressive treatments compared to older patients.”
Comments 3: From a surgical perspective, key information is missing that would help interpret the results. The authors should report treatment completion rates, reasons for discontinuation, nutritional status data (albumin, weight loss), performance status scores, and specific complications that led to reoperations. The finding that non-FLOT regimens led to more reoperations (22.2% vs 5.4%) is interesting but needs better explanation of what specific complications drove this difference.
Response 3: We do appreciate this comment. As reported in the previous Answer only patients who completed NACT were included in the NACT group. Please find the following statement added to the Materials and Methods section (page 3, line 124-125): “Only patients who completed NACT treatment were included in the NACT group for analysis.” Unfortunately, nutritional status data (albumin, weight loss) and performance status scores were not included in our dataset, despite their evaluation and influencing role on perioperative outcomes would have surely enriched the manuscript content. We agree with the reviewer that the higher rate of reoperation in the non-FLOT cohort is an interesting finding. This involved a total of 12 patients (8 in the non-FLOT group vs. 4 in the FLOT group), with highly heterogeneous causes for reoperation: 6 anastomotic leaks, 1 duodenal fistula, 3 cases of massive bleeding, 1 colonic fistula, and 1 case of small bowel ischemia. Further sub-analyses on such small subgroups, regarding either potential risk factors or specific complications, would be statistically underpowered and, thus, misleading for the reader.
Comments 4: The reassuring finding is the complication data showing similar rates of severe complications (20.9% vs 20.8%) and anastomotic leaks (12.7% vs 9.0%). This suggests that in appropriately selected patients, NACT doesn't significantly increase surgical risk, which matches clinical experience and is valuable information for practicing surgeons.
Response 4: Thank you for this positive observation. As correctly noted, the similar rates of severe complications and anastomotic leak between the NACT and upfront surgery groups suggest that, in appropriately selected patients, neoadjuvant chemotherapy does not significantly increase surgical risk. We believe this is an important and clinically relevant message, particularly for practicing surgeons involved in the multidisciplinary management of gastric cancer.
Comments 5: The authors need post-hoc power analysis for the negative survival findings to distinguish between "no effect" and "insufficient power." The 28% missing histotype data affects reliability of the Lauren classification analyses. The heterogeneous non-FLOT group (six different regimens plus "other") makes interpretation difficult. Short follow-up (41 months median) and 22% loss to follow-up introduce additional limitations. The authors should reposition this as a surgical safety analysis demonstrating that NACT is safe in selected patients rather than a comparative effectiveness study. The focus should be on what the data actually supports - the safety message is valuable and believable, while the survival equivalence claims are problematic due to selection bias.
Response 5: Thank you for the comment. We fully agree with the reviewer’s position. The data from our study are indeed affected by the heterogeneity of the non-FLOT group. Additionally, the limited follow-up time and the number of patients lost to follow-up represent further important limitations. However, we still considered it worthwhile to include these less conclusive findings, as they may serve as a useful starting point for future research. To better highlight the limitations of our study in evaluating overall survival (OS) and disease-free survival (DFS), we have added the following text to the Conclusions section (page 13, line 364-367): " Nevertheless, the considerable heterogeneity in chemotherapy regimens between the two cohorts, the relatively short follow-up period, and the substantial number of patients lost to follow-up limit the ability to draw definitive conclusions regarding the long-term impact of NACT and specific regimens on OS and DFS.”
Comments 6: The authors should acknowledge the selection bias prominently in the abstract and discussion. They need to explain why the survival results differ from published trials. Treatment completion rates and nutritional data should be included. Kaplan-Meier survival curves should be added. The manuscript needs professional English editing throughout.
Response 6: We do thank the reviewer for these observations, and we do agree on the need to better highlight the selection bias as the major limitation of our study. In order to better stress this point, as already reported in the Answers #3.1, 3.2 and 3.5, the Discussion section has been modified as follows (pag 12-13, line 332-347): “First, the relatively small sample size, and the retrospective study design and the presence of missing data (i.e. 72 cases with incomplete information on tumor histotype) significantly limit the generalizability of our results. Second, the lack of adequate matching between the two comparison groups in terms of clinico-demographic characteristics may have affected both perioperative and long-term outcome assessments; in particular, a potential limitation is that younger patients with advanced tumors may have received more aggressive treatments compared to older patients. Differences in tumor location also represent an important limitation of this study. In particular, patients with gastroesophageal junction tumors often require more complex and higher-risk surgical procedures, which are inherently associated with an increased likelihood of postoperative complications. Third, the subset analysis within the cohort of NACT patients is limited by the few numbers of individuals for type of non-FLOT regimens, affecting the reliability of the outcomes obtained. Furthermore, since non-FLOT protocols involve a variety of chemotherapeutic agents with differing toxicity profiles and efficacy, grouping them into a single analytical category may have intro-duced significant bias into the analysis.”
Limitations in the generalizability of our findings due to selection bias have been also reported in the Conclusions section of the Abstract, as follows (page 2, line 49-50):
“Although several selection biases limit the generalizability of our findings…”
Regarding the need to better specify the contrast between our data and the current available trials in terms of long-term outcomes after NACT, this has been underlined adding the following statement in the Discussion section (page 12, line 327-330):
“In our study, in contrast with the evidence reported by the MAGIC and FLOT4 trials[21,23], no significant difference was observed in 5-year OS between the NACT group (44.6%) and the upfront surgery group (47.7%) (p=0.96). These findings may be partially explained by several limitations inherent to our study.”
Moreover, in the subsequent paragraphs, we widely extended the limitations of our study in order to better report the current drawbacks that limit the generalizability of our results especially in terms of long-term outcomes (i.e. missing data, limited follow up, retrospective study design, limited sample size of the population).
Regarding the completion treatment rate, all patients in the NACT group completed the neoadjuvant treatment and treatment completion was an inclusion criterion. Please find the following statement added to the Materials and Methods section (page 3, line 124-125):
“Only patients who completed the planned NACT regimen were included in the NACT group for analysis.”
Conversely, no data on the nutritional status was present in our dataset. Kaplan-Meier curve for OS has been added as required and reported in the manuscript as Figure 1 (pag .
English was extensively revised by a mother-tongue English speaker
Comments 7: From a practical standpoint, what can clinicians tell patients based on this study? The safety data is reassuring for well-selected patients, but the survival data cannot be used to counsel patients about neoadjuvant benefits. The study is more useful as a safety analysis than an efficacy study.
Response 7: We do thank the reviewer for this keen observation, and we do agree on the need to better specify that no solid conclusions may be drawn in terms of long-term outcomes according to our findings, due to the presence of different study drawbacks. In this regard, please find the following statement added to the Conclusion paragraph:
“Nevertheless, the considerable heterogeneity in chemotherapy regimens between the two cohorts, the relatively short follow-up period, and the substantial number of patients lost to follow-up limit the ability to draw definitive conclusions regarding the long-term impact of NACT and specific regimens on OS and DFS”
Comments 8: This represents good institutional experience showing that NACT is surgically safe in appropriately selected patients. However, the survival conclusions are problematic due to selection bias. The paper would be stronger if repositioned as a surgical safety analysis rather than claiming treatment equivalence. With appropriate acknowledgment of limitations and focus on what the data actually demonstrates, this could make a valuable contribution to the literature. The manuscript needs major revision but has potential value for the surgical community.
Response 8: We do thank the reviewer for this conclusive evaluation of our study. Given the above-mentioned limitations, we do agree on the suggestion to focus more on the surgical safety analysis. In this regard, we extensively replied to all comments according to the reviewers’ observations, and we do hope the manuscript ameliorated both in content and readability. In addition, a more extensive description of the study limitations has been reported in order highlight the surgical perspective of the manuscript content and avoid any potential generalizability of the long-term outcomes.
|
Round 2
Reviewer 2 Report
Comments and Suggestions for Authors
The lack of novelty is the main reason for rejecting the paper to Cancers
Author Response
Comment 1: The lack of novelty is the main reason for rejecting the paper to Cancers
Response 1: We thank the reviewer for his consideration, and we do appreciate he judged the manuscript as well structured. With regards to the lack of novelty, we agree that previous studies on this topic are already available in the literature. However, we aimed to provide a further contribution to this topic by reporting a confirmatory experience from a high-volume referral center in the Western world.
Reviewer 3 Report
Comments and Suggestions for Authors
Thank you for the comprehensive revision addressing the concerns from the first review. The manuscript is substantially improved and now provides a valuable contribution to the literature on surgical safety of neoadjuvant chemotherapy in gastric cancer.
Minor Suggestions for Further Improvement
Line 46 (Abstract):
- Current: "while non-FLOT regimen was independently associated with an increased lower reoperation rate"
- Suggested: "while non-FLOT regimens were independently associated with higher reoperation rates"
Lines 49-50 (Abstract Conclusions):
- Current: "our results suggest that NACT prior to gastrectomy for GC does not increase postoperative morbidity and mortality"
- Suggested: "our results suggest that NACT prior to gastrectomy for GC does not increase postoperative morbidity and mortality in appropriately selected patients"
Lines 364-367 (Conclusions):
- Current: "limit the ability to draw definitive conclusions regarding the long-term impact of NACT and specific regimens on OS and DFS"
- Suggested: "prevent drawing definitive conclusions regarding the comparative long-term survival impact of NACT versus upfront surgery"
Author Response
Comment 1: Line 46 (Abstract):
Current: "while non-FLOT regimen was independently associated with an increased lower reoperation rate"
Suggested: "while non-FLOT regimens were independently associated with higher reoperation rates"
Answer 1: We thank for the suggestions. Modifications have been done as required.
Comment 2: Lines 49-50 (Abstract Conclusions):
Current: "our results suggest that NACT prior to gastrectomy for GC does not increase postoperative morbidity and mortality"
Suggested: "our results suggest that NACT prior to gastrectomy for GC does not increase postoperative morbidity and mortality in appropriately selected patients"
Answer 2: We do appreciate the suggestion. We added “in appropriately selected patients” in the Conclusion paragraph of the Abstract.
Comment 3: Lines 364-367 (Conclusions):
Current: "limit the ability to draw definitive conclusions regarding the long-term impact of NACT and specific regimens on OS and DFS"
Suggested: "prevent drawing definitive conclusions regarding the comparative long-term survival impact of NACT versus upfront surgery"
Answer: Thank you for the suggestion. The text was modified accordingly